# Characterization of Triplet State of Cyanine Dyes with Two Chromophores Effect of Molecule Structure

**Iouri E. Borissevitch** [1,*] **, Pablo J. Gonçalves** [2] **, Lucimara P. Ferreira** [3] **, Alexey A. Kostyukov** [4] **and Vladimir A. Kuzmin** [4]

1    Departamento de Fisica, Faculdade de Filosofia, Ciencias e Letras de Ribeirao Preto, Universidade de Sao Paulo, Ribeirao Preto 14040-901, Brazil
2    Instituto de Fisica, Universidade Federal de Goias, Goiania 74001-970, Brazil; pablaojg@yahoo.com.br
3    Centro Universitário da Fundação Educacional de Barretos (UNIFEB), Barretos 14783-226, Brazil; lucimaraferreira@yahoo.com.br
4    N. M. Emanuel Institute of Biochemical Physics, Russian Academy of Sciences, 119334 Moscow, Russia; akostyukov@gmail.com (A.A.K.); vladimirkuzmin7@gmail.com (V.A.K.)
*    Correspondence: iourib@ffclrp.usp.br

**Abstract:** Quantum yields ($\varphi_T$) and energies ($E_T$) of the first triplet state $T_1$ for four molecules of cyanine dyes with two chromophores (BCDs), promising photoactive compounds for various applications, for example, as photosensitizers in photodynamic therapy (PDT) and fluorescence diagnostics (FD), were studied in 1-propanol solutions by steady-state and time-resolved optical absorption techniques. BCDs differ by the structure of the central heterocycle, connecting the chromophores. The heterocycle structure is responsible for electron tunneling between chromophores, for which efficiency can be characterized by splitting of the BCD triplet energy levels. It was shown that the increase in the tunneling efficiency reduces $E_T$ values and increases $\varphi_T$ values. This aspect is very promising for the synthesis of new effective photosensitizers based on cyanine dyes with two interacting chromophores for various applications, including photodynamic therapy.

**Keywords:** bichromophoric cyanine dyes; triplet state quantum yield; triplet state energy; electron tunneling; central heterocycle structure; photodynamic therapy

## 1. Introduction

The purpose of this study was to attempt to establish the relationship between the structure of cyanine dye molecules with two interacting chromophores and the characteristics of their first triplet state $T_1$, namely, its quantum yield ($\varphi_T$) and energy ($E_T$).

Cyanine dyes (CDs) are organic compounds with spectral characteristics determined by a linear polymethine π-conjugated chain (chromophore) in their structure [1,2]. CD absorption spectra are characterized by intensive and narrow optical absorption bands, whose position depends on the length of the π-conjugated chain and which can be localized in the spectral range from near ultraviolet up to near infrared [1,2]. CDs possess affinity to biological structures, such as proteins, nucleic acids, and cell membranes [1,3]. For CDs, the process of photoisomerization of the polymethine chain is typical [1,2]. The reverse isomerization of photoisomers to the initial isomeric state of the dye occurs in the dark and is one of the types of nonradiative dissipation of the CD excited state energy, which dramatically reduces lifetimes and quantum yields of the CD fluorescence ($\tau_{fl}$ and $\varphi_{fl}$) and excited triplet state $T_1$ ($\tau_T$ and $\varphi_T$). However, when bound with biological structures, CDs demonstrate relatively high $\varphi_{fl}$ and $\varphi_T$ [3–5]; this effect is associated with increasing the CD structure rigidity due to binding with more rigid structures. Because of their intense fluorescence and affinity for biological structures, CDs are widely applied in biology and medicine as fluorescent probes (FPs) [1–5].

The photosensitizer triplet state plays an important role in photodynamic therapy (PDT) since PDT mechanisms are associated with the formation of reactive oxygen species (ROS), such as (1) molecular oxygen in its singlet excited state ("singlet oxygen"), formed via energy transfer from the photosensitizer triplet state to molecular oxygen in the ground state (PDT type II), and (2) radical species, anion superoxide radicals, in particular, formed via electron transfer from the photosensitizer triplet state to molecular oxygen (PDT type I). Therefore, the fact that CDs bound to biological structures possess high $\varphi_T$ values makes them promising photosensitizers (PSs) in PDT [6]. To be applicable in PDT, PSs should possess optical absorption in the spectral range 600 nm < λ < 800 nm ("phototherapeutic window"), where biological tissues are relatively transparent. CDs with optical absorption in this region have a long polymethine π-conjugated chain [1–3]. This makes them more flexible. Moreover, increasing the π-conjugated chain length increases the probability of CD photoisomerization. Both these effects increase the contribution of nonradiative mechanism to the energy dissipation of the CD's electronic $S_1$ and $T_1$ excited states, thus reducing the CD's $\varphi_{fl}$ and $\varphi_T$ [5,6] and, consequently, its efficiency as a PS for PDT and as an FP for photodiagnostics.

Absorption spectra of CDs with two chromophores (bichromophoric cyanine dyes, BCDs) possess an intensive optical absorption with the molar absorption coefficient at the absorption maximum $\approx 10^5$ $M^{-1}cm^{-1}$. BCD absorption spectra [7,8] undergo redshift not only due to the increase in the chromophore chain lengths, but mainly due to interaction between chromophores, which includes two effects: dipole–dipole interaction between chromophores [8] and electron tunneling through the central heterocycle [9,10]. Therefore, BCD molecules with absorption peaks in the spectral region of the phototherapeutic window are less flexible, as compared with a single-chromophore CD. This is confirmed by the fact that BCD fluorescence and $T_1$ states have already been observed in homogeneous solutions [11,12]. High efficiency of $T_1$ state formation is responsible for BCD high photocytotoxicity toward cancer cells [13], which makes them promising as PSs for PDT. One more fact which confirms the effective formation of the $T_1$ state at BCD photoexcitation is that BCDs suffer phototransformation in the presence of molecular oxygen [14]. This process is probably due to the reaction of BCD molecules either with the singlet oxygen formed via energy transfer from BCD molecules in the $T_1$ state to oxygen molecules, or with anion superoxide radicals formed via electron transfer from the BCD $T_1$ state to the oxygen molecule.

The profiles of the absorption spectra of BCD ground and $T_1$ states depend on the structure of the central heterocycle, which couples the chromophores, determining the angle and effective distance between them and the effectivity of electron tunneling through the heterocycle [7–9,15]. Therefore, it is reasonable to expect that other $T_1$ state characteristics should also depend on its structure.

Among others, quantum yield, $\varphi_T$, and energy, $E_T$, are important parameters of the triplet state, which help to characterize its reactivity. In this work, we present the results of $E_T$ and $\varphi_T$ studies for four BCDs with different structures of the central heterocycle. The study was carried out in 1-propanol solutions using steady-state and time-resolved absorption spectroscopy. It was demonstrated that the increase in the tunneling efficiency reduces $E_T$ values and increases $\varphi_T$ values, thus increasing the efficacy of BCD in PDT. One can expect that BCDs with the central heterocycle possessing major efficiency of the electron tunneling will possess higher $\varphi_T$. This aspect is very promising for the synthesis of new effective PSs based on cyanine dyes with two interacting chromophores.

## 2. Materials and Methods

The bichromophoric cyanine dyes (BCDs, Figure 1) were synthesized at the Institute of Organic Chemistry, National Academy of Sciences of Ukraine. A detailed description of the dye synthesis and analysis of their purity is presented in [16]. These dyes were obtained from the collection of GOSNIIKHIMPHOTOPROEKT Company (Moscow, Russia). The dye solutions were prepared in 1-propanol obtained from Sigma-Aldrich Company.

**BCD1**

**BCD2**

**BCD3**

**BCD4**

**Figure 1.** Structures of bichromophoric cyanine dyes (BCDs).

Quantum yields ($\varphi_T$) of the BCD $T_1$ state were determined by a relative method, described in [17], using *meso*-tetrakis(p-sulfonatofenyl) porphyrin (TPPS$_4$) in water at pH 4.0 ($\varphi_T^s = 0.36$, [18]) as a standard. TPPS$_4$ was purchased from Porphyrin Products Inc. The

porphyrin solutions were prepared in milli-Q quality water. The pH value was adjusted by adding aliquots of HCl stock solution.

The BCD $T_1$ state energies ($E_T$) were defined in relation to that of azulene, purchased from Porphyrin Products Inc. The experiments were carried out in the temperature interval from 278 K up to 303 K with 5° steps. The sample temperature was controlled by a Copper/Constantan thermocouple. The temperature measurement accuracy was ±1°.

Since interaction with molecular oxygen reduces the triplet state lifetime due to the energy or electron transfer between molecules in the triplet state and oxygen [19,20], all experiments were carried out in de-aerated solutions at room temperature (297 K). De-aeration of the solution was achieved by bubbling nitrogen through the experimental cell for 20 min.

The sample optical absorption spectra were monitored by a Beckman Coulter DU-640 spectrophotometer.

Excited triplet states ($T_1$) were analyzed using the Laser Flash Photolysis (LFP) technique [21]. The $T_1$ states of both TPPS$_4$ and BCDs were obtained by excitation of their solutions by the third harmonics (355 nm) of a Nd:YAG laser (Quantel, model Brio, Les Ulis, France) with 10 Hz pulse repetition frequency and 5 ns pulse duration. The analyzing light beam was produced by light source with a 75 W Xenon lamp (XE075), a quartz collimator, a light chopper, and a monochromator Sciencetech Inc (9055) coupled to a Hamamatsu Photomultiplier tube (R928). The analyzing light beam direction was orthogonal to the pump beam. The $T_1$ state decay curves were monitored by the triplet–triplet absorption at 470 nm for TPPS$_4$ and at 670 nm for the BCD. The measurements were carried out in a 1 cm × 1 cm quartz spectroscopic cell.

The experimental data were treated using the OriginPro 8 commercial program. All final values were averages, obtained in three independent experiments.

### 2.1. Determination of the $T_1$ State Quantum Yield

Initially, in the thermodynamic equilibrium, practically all the dye molecules are in the ground state $S_0$. After absorption of a photon, the molecule obtains excessive electronic energy, passing to the first singlet excited state $S_1$. The excess of $S_1$ state energy may dissipate via three ways: fluorescence and internal conversion, through which the molecule returns to its initial $S_0$ state, and intersystem crossing, due to which the molecule passes to the excited triplet state $T_1$. Competition between these ways determines the $T_1$ state quantum yield, defined as follows:

$$\varphi_T = \frac{n_{T1}}{n_{abs}}$$

where $n_{abs}$ is the number of photons absorbed by the sample and $n_{T1}$ is the number of molecules in the triplet state, formed by the absorption of these photons. The $\varphi_T$ value can be determined in two ways: (1) using the analysis of the kinetic characteristics of the $S_1$ state energy dissipation; or (2) using the spectroscopic characteristics of the system after its excitation by light.

In this study we used the second way. The procedure of the quantum yield $\varphi_T$ determination via this way was described in detail in [17]. The $\varphi_T$ was calculated in accordance with the following equation:

$$\varphi_T = \frac{\Delta A_0}{\Delta A_0^s} \frac{A_{ex}^s}{A_{ex}} \frac{C_0}{C_0^s} \frac{\Delta A_{max}^s}{\Delta A_{max}} \varphi_T^s \tag{1}$$

where $\Delta A_0$ and $\Delta A_0^s$, are amplitudes of the triplet state decay curves of the sample and the standard, measured immediately after the end of the exciting pulse (Figure 2A), $A_{ex}$ and $A_{ex}^s$ are absorbances of the sample and the standard solutions at the excitation wavelength, $C_0$ and $C_0^s$ are concentrations of the sample and the standard solutions, $\Delta A_{max}$ and $\Delta A_{max}^s$ are the maximum values of $\Delta A_0$ and $\Delta A_0^s$, obtained from approximation of $\Delta A_0$ and $\Delta A_0^s$

dependences on the exciting pulse energy $J$ for $J \to \infty$, and $\varphi_T^s$ is the quantum yield of the triplet state of the standard. Since the $T_1$ state lifetimes for the BCD and porphyrin in the absence of oxygen are in the order of hundreds of microseconds and the exciting pulse duration is 5 ns, the process of $T_1$ state deactivation during the pulse is negligible and cannot affect $\Delta A_0$ and $\Delta A_0^s$.

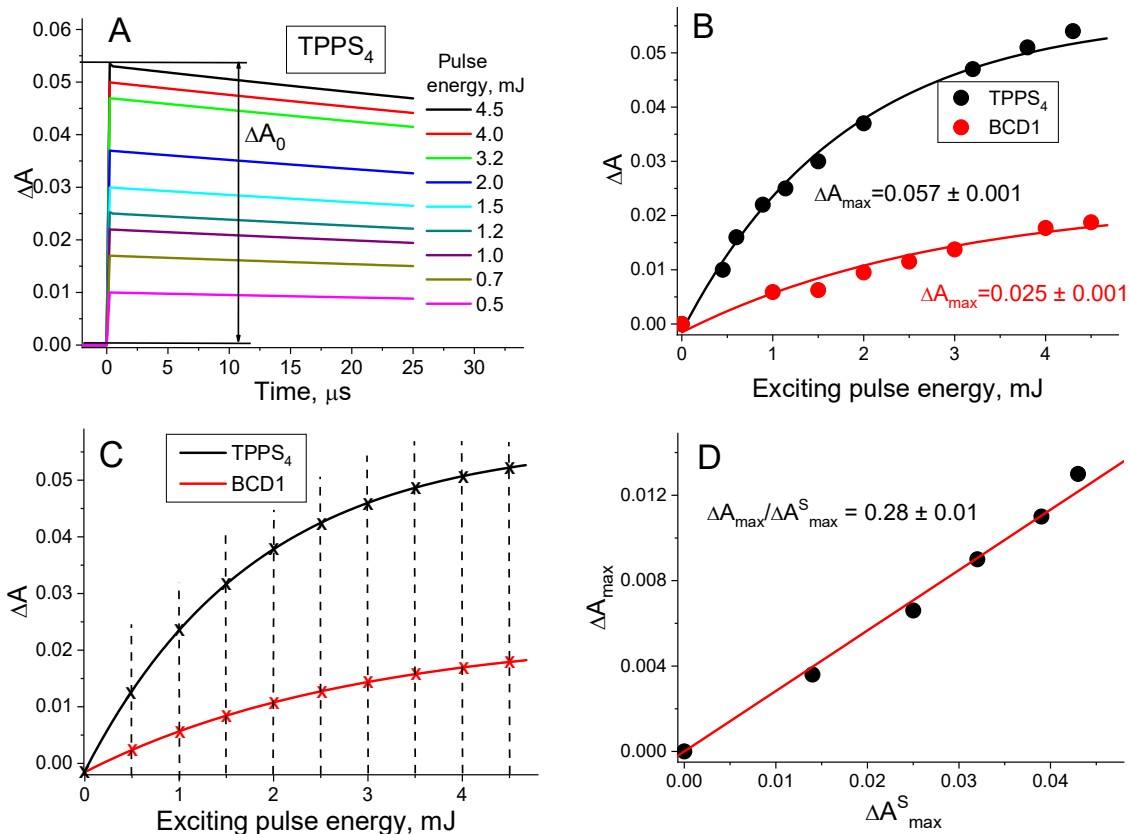

**Figure 2.** Determination of the BCD1 $T_1$ state quantum yield. (**A**) $T_1$ state decay curves for various exciting pulse energies (*E*, mJ). for the standard (TPPS$_4$). (**B**) Dependences of the amplitudes of $T_1$ state decay curves ($\Delta A_{0max}$) on *E* for the standard ($\lambda = 470$ nm) and BCD1 ($\lambda = 550$ nm) and relative fittings. (**C**) Definition of $\Delta A_{0max}$ of the standard and BCD 1 for the same E from the experimental data fitting. (**D**) Dependence of the BCD1 $\Delta A_{0max}$ on that of the standard ($\Delta A_{0max}^S$).

All the values in this equation, except $\Delta A_0^s$, are independent of the exciting pulse energy. Thus, by varying the exciting pulse energy we can obtain the $\Delta A_0$ values in the function of $\Delta A_0^s$, and determine $\varphi_T$ from the dependence of $\Delta A$ on $\Delta A^s$.

### 2.2. Determination of the BCD $T_1$ State Energy

To determine the energy of the BCD $T_1$ state, the reaction of the triplet–singlet energy transfer from the donor in its triplet state ($D(T_1^*)$) to the acceptor in the ground state ($A(S_0)$) was applied [22].

$$D(T_1^*) + A(S_0) \to D(S_0) + A(T_1^*)$$

The triplet–singlet energy transfer is realized via the Dexter mechanism of simultaneous exchange of electrons between excited and ground states [22,23].

To reduce the probability of the reverse energy transfer from $A(T_1^*)$ to $D(S_0)$ an acceptor with an extremely short $T_1^*$ lifetime was chosen.

In the absence of the reverse energy transfer, the time dependence of the donor triplet state concentration $[D(T_1^*)]$ after the end of the exciting light pulse is determined by the following equation:

$$\frac{d[D(T_1^*)]}{dt} = -k_0[D(T_1^*)] - k_q[D(T_1^*)][A(S_0)] \tag{2}$$

where $k_0$ is the constant of the $D(T_1^*)$ state deactivation in the absence of the acceptor, $k_q$ is the constant of the $D(T_1^*)$ state bimolecular quenching by the acceptor, and $[A(S_0)]$ is the acceptor concentration.

Since $[A(S_0)] >> [D(T_1^*)]$, the $[A(S_0)]$ can be considered constant and the $[D(T_1^*)]$ time dependence can be expressed as follows:

$$[D(T_1^*)] = [D(T_1^*)]_0 exp\{-(k_0 + k_q[A(S_0)])t\} \tag{3}$$

where $[D(T_1^*)]_0$ is the donor $T_1^*$ state concentration immediately after the end of the exciting pulse.

At the wavelengths, where the acceptor possesses no optical absorption, the sample absorbance is as follows:

$$A = A_S + A_T = \varepsilon_S[D(S_0)] + \varepsilon_T[D(T_1^*)] \tag{4}$$

where $A_S$ and $A_T$ are absorbances of the donor molecules in the ground (singlet) state and in the excited (triplet) state, respectively, and $\varepsilon_S$ and $\varepsilon_T$ are molar absorption coefficients of the ground and the triplet states.

Since the sum of concentrations in the ground and the excited states is equal to the initial compound concentration $[D(S_0)] + [D(T_1^*)] = [D(S_0)]_0$, Equation (4) can be rewritten as follows:

$$A = \varepsilon_S([D(S_0)]_0 - [D(T_1^*)]) + \varepsilon_T[D(T_1^*)] = A_0 + (\varepsilon_T - \varepsilon_S)[D(T_1^*)] \tag{5}$$

where $A_0$ is the solution absorbance before the exciting pulse action.

Thus, the absorbance change after the pulse action can be expressed as follows:

$$\Delta A = A_0 - A = (\varepsilon_S - \varepsilon_T)[D(T_1^*)] \tag{5a}$$

and the concentration of molecules in the triplet state can be expressed as follows:

$$[D(T_1^*)] = \frac{\Delta A}{\varepsilon_S - \varepsilon_T} \tag{5b}$$

Thus, Equation (3) can be rewritten as follows:

$$\Delta A = \Delta A_0 exp\{-(k_0 + k_q[A(S_0)])t\} \tag{6}$$

and

$$ln\frac{\Delta A_0}{\Delta A} = (k_0 + k_q[A(S_0)])t \tag{7}$$

The $k_q$ value can be obtained as the slope of the graph of $ln\frac{\Delta A_0}{\Delta A}$ as a function of $[A(S_0)]$.

When the acceptor $T_1^*$ state energy is lower than that of the donor, the energy transfer occurs at every contact of the excited donor molecule with the acceptor and the $k_q$ value is characteristic for the diffusion-controlled process. However, when the acceptor triplet state energy is higher than that of the donor, thermic activation energy ($E_{act}$) is necessary to realize the energy transfer (Scheme 1).

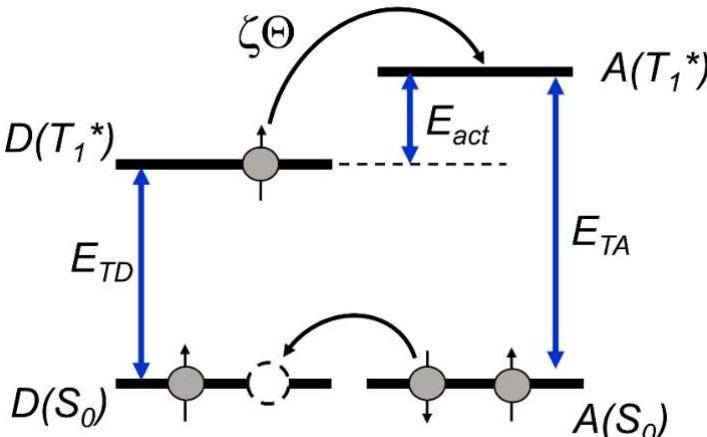

**Scheme 1.** Energy transfer from the donor in the triplet state to the acceptor via thermal activation; $E_{act}$ is the activation energy, and $E_{TD}$ and $E_{TA}$ are the donor and acceptor triplet energies, respectively.

In this case $k_q$ should be expressed as follows:

$$k_q = k_{q\infty} exp\left(-\frac{E_{act}}{\zeta\Theta}\right) \tag{8}$$

where $\zeta$ is the Boltzmann constant, $\Theta$ is the temperature in Kelvin, and $k_{q\infty}$ is the bimolecular quenching constant at $\Theta \to \infty$.

Thus

$$lnk_q = lnk_{q\infty} - \frac{E_{act}}{\zeta\Theta} \tag{9}$$

The $E_{act}$ value can be obtained as the slope of the graph of $lnk_q$ as a function of $\frac{1}{\Theta}$ and the donor triplet energy ($E_{TD}$) can be calculated as follows:

$$E_{TD} = E_{TA} - E_{act} \tag{10}$$

Equation (10) was applied to determine the triplet energy of the BCD ($E_T^{BCD}$) used as the triplet energy donor toward azulene acting as the acceptor with $E_{TA} = 13,600 \text{ cm}^{-1}$ [24]. The azulene $T_1^*$ lifetime is $\approx 0.1$ μs [25], which practically excludes the reverse energy transfer from its triplet state to the BCD.

## 3. Results and Discussion

### 3.1. Determination of the BCD $T_1$ State Quantum Yields ($\varphi_T$)

The process of the $\varphi_T$ determination for BCD 1, as an example, is illustrated in Figure 2A–D.

1.  In the first step, the amplitudes of the $T_1$ state decay kinetic curves of the BCD and of the standard compound ($\Delta A_0$ and $\Delta A_0^S$) were measured for various exciting pulse energies ($E$) (Figure 2A).
2.  In the second step, the dependences of $\Delta A_0$ and $\Delta A_0^S$ on $E$ were fitted in accordance with the mono-exponential equation $\Delta A_0 = \Delta A_{0max} - \Delta A_{0max} \times \exp\left(-\frac{E}{E_{2.3}}\right)$ (Figure 2B). From this fitting, the $\Delta A_{0max}$ and $\Delta A_{0max}^S$ values were obtained.
3.  In the third step, the obtained fittings were used to determine $\Delta A_0$ and $\Delta A_0^S$ for the same $E$ values (Figure 2C).
4.  In the fourth step, the dependence of $\Delta A_0$ on $\Delta A_0^S$ was constructed to determine the $\frac{\Delta A_0}{\Delta A_0^S}$ average value (Figure 2D).

The obtained values were used for calculation of $\varphi_T^{BCD1}$ in accordance with Equation (1).

### 3.2. Determination of the BCD $T_1$ State Energy ($E_T$)

The process of the $E_T$ determination for BCD 1 is illustrated in Figure 3A,B.

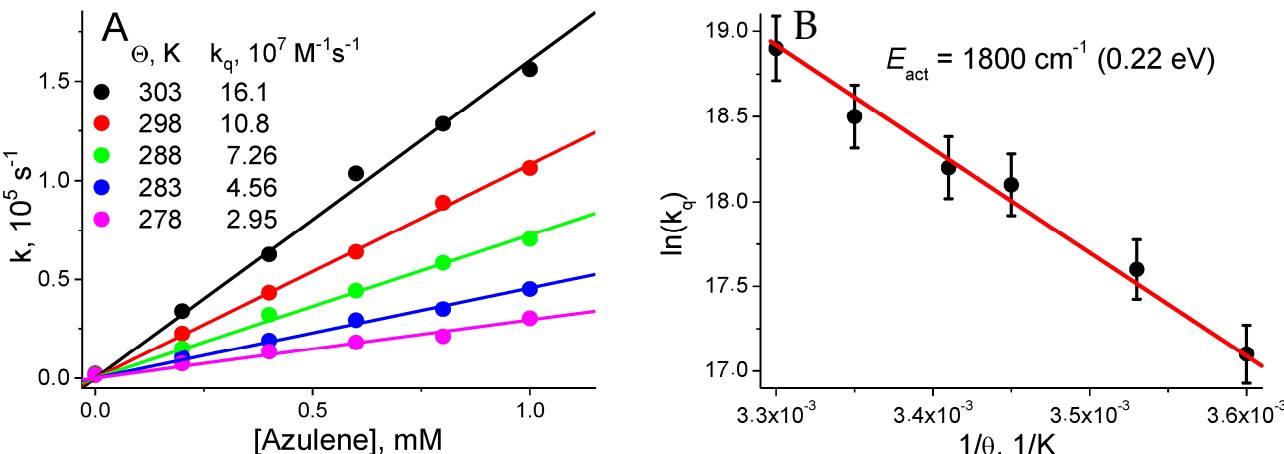

**Figure 3.** (**A**) Dependences of the BCD1 $T_1$ state decay constants ($k$, $s^{-1}$) as a function of azulene concentration at different temperatures. (**B**) Dependence of logarithm of the BCD $T_1$ state quenching constants ($k_q$, $M^{-1}s^{-1}$) as a function of the reverse absolute temperature ($1/\Theta$, $1/K$).

1. In the first step, the quenching constants of the BCD $T_1$ state by azulene ($k_q$) at different temperatures were determined (Figure 3A).
2. In the second step, the dependence of $\ln(k_q)$ on the reverse absolute temperature ($1/\Theta$) was constructed to determine the activation energy ($E_{act}$), which is necessary to realize the energy transfer from the BCD $T_1$ state to azulene (Figure 3B).

Since the azulene $T_1$ state energy $E_{TA} = 13{,}600\ cm^{-1}$, the BCD1 $T_1$ state energy is

$$E_T^{BCD1} = 13{,}600\ cm^{-1} - 1800\ cm^{-1} = 11{,}800\ cm^{-1}$$

The triplet state characteristics for four BCDs are summarized in Table 1.

**Table 1.** The angle between BCD chromophores $\alpha$, BCD $T_1$ state quantum yields ($\varphi_T$), the energies of the lowest $T_1$ and the $S_1$ state levels ($E_T$ and $E_S$, respectively), splitting of the triplet energy levels ($\Delta E_T$), and the difference in $S_1$ and $T_1$ level energy ($\Delta E_{S-T}$).

| Dye | $\alpha$ | $\varphi_T$ | $E_T$, cm$^{-1}$ | $\Delta E_T$, cm$^{-1}$ [8] | $E_S$, cm$^{-1}$ [7] | $\Delta E_{S-T}$, cm$^{-1}$ |
|---|---|---|---|---|---|---|
| **BCD1** | 180 | $0.21 \pm 0.02$ | $11{,}800 \pm 200$ | $3700 \pm 100$ | $15{,}270 \pm 10$ | $3500 \pm 100$ |
| **BCD2** | $151 \pm 2$ | $0.18 \pm 0.02$ | $12{,}000 \pm 200$ | $3600 \pm 100$ | $15{,}630 \pm 10$ | $3600 \pm 100$ |
| **BCD3** | $91 \pm 1$ | $0.08 \pm 0.02$ | $12{,}400 \pm 300$ | $3500 \pm 100$ | $15{,}720 \pm 10$ | $3300 \pm 100$ |
| **BCD4** | $123 \pm 4$ | $0.04 \pm 0.03$ | $12{,}900 \pm 400$ | $3300 \pm 100$ | $16{,}530 \pm 10$ | $3600 \pm 100$ |

### 3.3. Discussion

The interaction between chromophores in the BCD molecule with two identical chromophores induces splitting of the singlet energy level of the respective dye with an adequate single chromophore. This splitting is responsible for two bands in the BCD singlet–singlet absorption spectrum, red- and blueshifted as compared with the respective CD with a single chromophore [7,8]. Relative intensities of these two absorption bands depend on the angle $\alpha$ between chromophores. So, when $\alpha = 180°$, just the long-wavelength band is present in the dye absorption spectrum, whereas when $\alpha = 0°$, just the short-wavelength band appears in the spectrum, and at $\alpha = 90°$ the intensities of both bands are practically equal [7,8] (Table 2).

**Table 2.** Positions of the maxima of BCD optical absorption bands ($\lambda_1$ and $\lambda_2$) and respective molar absorption coefficients ($\varepsilon_1$ and $\varepsilon_2$).

| Dye | $\alpha^\circ$ | $\lambda_1$, nm | $\varepsilon_1 \times 10^{-5}$, $M^{-1}cm^{-1}$ | $\lambda_2$, nm | $\varepsilon_2 \times 10^{-5}$, $M^{-1}cm^{-1}$ |
|---|---|---|---|---|---|
| BCD1 | 180 | -- | -- | 655 | 2.8 |
| BCD2 | $151 \pm 2$ | 525 | 0.2 | 640 | 2.4 |
| BCD3 | $91 \pm 1$ | 520 | 1.5 | 634 | 1.3 |
| BCD4 | $123 \pm 4$ | 525 | 0.6 | 605 | 2.1 |

Initially this effect was explained by A. I. Kiprianov and G. G. Dyadyusha as provoked by the dipole–dipole interaction between chromophores within the framework of the theory of molecular dipoles [26]. Based on this theory, they derived an equation that allows the angle $\alpha$ between the chromophores in the BCD molecule to be calculated using the characteristics of the absorption spectrum of the dye [8]:

$$\cos\alpha = \frac{\lambda_1\varepsilon_1/\lambda_2\varepsilon_2 - 1}{\lambda_1\varepsilon_1/\lambda_2\varepsilon_2 + 1} \qquad (11)$$

where $\lambda_1$ and $\lambda_2$ are wavelengths of the maxima and $\varepsilon_1$ and $\varepsilon_2$ are molar absorption coefficients of the short-wavelength and long-wavelength absorption bands, respectively.

The validity of this equation was confirmed by crystallographic studies and quantum chemical calculations [10].

However, later it was found that, in addition to the dipole–dipole interaction, the splitting of the singlet energy levels of the BCD molecule may also be due to electron tunneling through the central heterocycle that connects the chromophores [9,10]. Moreover, in accordance with the theory [27,28], the dipole–dipole interaction cannot produce splitting of the triplet energy levels. Therefore, the observed line splitting in the triplet–triplet absorption spectrum of the BCD molecule is only due to the electron tunneling through the central heterocycle.

An important fact is that Equation (11) is valid for both mechanisms of the chromophore interaction, since relative absorption band intensities depend only on the spatial configuration of the molecule.

The splitting of the singlet energy level produced by the dipole–dipole interaction is symmetric in energy units in relation to the singlet energy level for the respective single-chromophore CD. The structure of the BCD central heterocycle determines the angle between chromophores (Figure 1), thus modifying the distance between them and, consequently, changing the degree of the dipole–dipole interaction and the magnitude of the splitting of the singlet levels.

At the same time, the splitting of the BCD singlet energy levels produced by the electron tunneling through the central heterocycle is asymmetric towards the singlet energy level of the respective CD, and both new levels possess energies lower than the energy level of the respective CD. The structure of the BCD central heterocycle determines the efficiency of the electron tunneling and, consequently, the splitting values [9,10]. Therefore, the total value of the splitting of the BCD singlet energy level, which is the sum of those produced both by the dipole–dipole interaction and the electron tunneling, depends on the structure of the central heterocycle of the BCD molecule.

Since the dipole–dipole interaction does not contribute to the splitting of the triplet energy levels, which occurs only due to the electron tunneling through the central heterocycle of the molecule, the values of the splitting of the triplet–triplet absorption spectrum $\Delta E_T$ can be used to characterize the effectiveness of the electron tunneling in the set of similar BCDs. The $\Delta E_T$ values increase in the sequence BCD4 < BCD3 < BCD2 < BCD1 (Table 1). Thus, it is possible to conclude that the tunneling effect increases in the same sequence.

Moreover, the comparison of $\varphi_T$ and $E_T$ values with $\Delta E_T$ (Table 1) demonstrates that the increase in the electron tunneling efficiency increases $\varphi_T$ and reduces $E_T$.

One could assume that the difference in $\varphi_T$ is due to the difference in the $S_1$ and the $T_1$ state energies. However, the analysis of $S_1$-$T_1$ splitting ($\Delta E_{S-T}$) shows that $\Delta E_{S-T}$ values are close for all the BCDs (Table 1). Therefore, it is necessary to look for another explanation for the effect of the BCD structure upon $\varphi_T$.

As it was demonstrated formerly, the quantum yield of the triplet state of a single-chromophore CD in homogeneous liquid solutions with low viscosity (ethanol, 1-propanol) is extremely low. However, it increases with the increase in the solvent viscosity and can be clearly observed, for example, in glycerol. As mentioned above [29], the same effect was observed for CD bound with biological and model nanostructures, such as proteins, DNA, and micelles [30]. Similar behavior was observed for the quantum yield of the CD fluorescence, which increases with the solvent viscosity and at the dye binding with rigid structures. So, for example, the quantum yield of the BCD1 fluorescence for the dye bound with DNA increases by more than 20 times, as compared with a homogeneous solution [31]. The opposite effect of viscosity was observed for the CD photoisomerization [30]. The increase in viscosity or binding of the dye with a rigid structure reduces the quantum yield of the CD photoisomerization. All these facts show that low quantum yield of the triplet state of a single-chromophore CD is a result of the effective deactivation of its singlet excited state via nonradiative processes: interconversion and re-isomerization of the photo isomers to the initial isomeric state of the dye. The increase in the rigidity of the dye molecule due to the increase in the solvent viscosity or to the dye binding with a rigid structure reduces the probability of nonradiative processes of deactivation of the dye singlet excited state energy in favor of the fluorescence and the triplet state formation.

Differently from a single-chromophore CD, in the case of a BCD with the same length of the chromophore $\pi$-conjugated chain, we observe no photoisomerization of the BCD chromophores, which are in trans configuration for the BCD in the ground state [10]. Therefore, we can conclude that the structure of cyanine dyes with two chromophores does not favor photoisomerization, thus excluding the canal of dissipation of their singlet excited state energy via re-isomerization. The increase in the BCD triplet state quantum yield with the increase in the electron tunneling efficiency ($\Delta E_T$ increase, Table 1) shows that electron tunneling increases the rigidity of the chromophore $\pi$-conjugated chain, thus reducing the probability of nonradiative processes of the singlet excited state energy dissipation and increasing the probability of the triplet state formation.

The absence of correlation between the BCD $\varphi_T$ values and the angles between chromophores and, consequently, with the distance between them, shows that the dipole–dipole interaction between chromophores does not contribute to nonradiative processes of the excited state energy dissipation.

The reduction in the triplet state energy $E_T$ with the increase in $\Delta E_T$ is clearly associated with the increase in the $T_1$ state level splitting when the electron tunneling efficiency increases.

Thus, using heterocycles with structures providing different efficiencies of the electron tunneling, it is possible to synthesize BCD with higher or lower quantum yield and the energy of its triplet state.

BCDs possess intensive optical absorption in the spectral range $\lambda > 600$ nm (region of the phototherapeutic window), high affinity with biological structures, and relatively high $\varphi_T$. We have already shown that BCD1 demonstrates higher photocytotoxicity toward cancer cells than Photofrin compound, which is applied nowadays in clinics as the PS in PDT cancer treatment [13], and, independently, that BCD1 $\varphi_T$ is much lower than that of Photofrin. It was shown that its penetration into the cell interior is much faster than that of Photofrin [13]. Thus, we can consider BCD as a promising PS for PDT.

Generally, only compounds with the triplet state quantum yield > 50% are considered as promising for PDT [32]. However, it is valid for PS molecules with rigid structures, such as porphyrin-like compounds. The $\varphi_T$ values for these compounds demonstrate low dependence on the molecule environment, while for CDs, $\varphi_T$ increases dramatically at

the dye binding with biological structures. We expect the same effect for a BCD, which could explain its high photocytotoxicity. This supposition is in accordance with the fact that the quantum yield of the BCD 1 fluorescence increases more than 20 times (from 0.02 to 0.44) [31] when the dye is bound with DNA. Moreover, one can expect that the BCD with the central heterocycle possessing major efficiency of the electron tunneling will possess higher $\varphi_T$. This aspect is very promising from the point of view of the synthesis of new effective PS, but it needs more detailed studies.

### 4. Conclusions

Tunneling of electrons through the central heterocycle in cyanine dyes with two chromophores induces splitting of their triplet state energy levels. Therefore, the splitting value can be used to characterize the tunneling efficiency. The increase in the tunneling efficiency and, consequently, the increase in the splitting, produces the reduction in the energy of the BCD triplet state. In addition, the electron tunneling stabilizes the BCD chromophore structure, reducing the contribution of nonradiative processes in the dissipation of the singlet excited state energy and increasing the quantum yield of the first BCD triplet state, thus increasing the efficacy of BCD in PDT. Thus, one can expect that a BCD with the central heterocycle possessing major efficiency of the electron tunneling will possess higher $\varphi_T$. This aspect is very promising for the synthesis of new effective PSs based on cyanine dyes with two interacting chromophores.

**Author Contributions:** I.E.B.—initial idea, general coordination of the participants, experiment execution, analysis of the results and discussion, preparation of the text; P.J.G.—experiment execution, analysis of the results and discussion, preparation of the text; L.P.F.—experiment execution, analysis of the results and discussion; A.A.K.—analysis of the results and discussion, correction of the text; V.A.K.—discussion of the initial idea, analysis of the results and discussion, correction of the text. All authors have read and agreed to the published version of the manuscript.

**Funding:** This research was supported by the Brazilian funding agencies: Conselho Nacional de Desenvolvimento Científico e Tecnológico (CNPq) Grants No. 303764/2021-0 and 310200/2021-0), Fundação de Amparo à Pesquisa do Estado de Goiás (FAPEG)—Grants No. 201410267001776 and 201710267000533); and Coordenação de Aperfeiçoamento de Pessoal de Nível Superior (CAPES) Finance Code 001.

**Institutional Review Board Statement:** This study did not require ethical approval.

**Informed Consent Statement:** This study did not include the studies with human beings.

**Data Availability Statement:** All the data supporting reported results can be found in the list of references.

**Conflicts of Interest:** The authors declare no conflict of interest.

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
