# Peer review of "Characterization of Triplet State of Cyanine Dyes with Two Chromophores Effect of Molecule Structure"

_technologies, doi:10.3390/technologies11040090_

Round 1

Reviewer 1 Report

Comments and Suggestions for Authors

The manuscript describes a study of the first triplet state, quantum yields and energies, of four bichromophoric cyanine dyes (BCD) in 1-propanol by Laser Flash Photolysis. Quantum yields of the T1 state were determined by a relative method using porphyrin (TPPS4), and the T1 state energies were determined in relation to azulene.

The topic is interesting and relevant, and the manuscript is mostly well written. However, while this paper fits well into the Quantum Technologies section of the journal, it should show greater importance for the technology it relates to. The final claim in the abstract and the conclusions, that the described ‘effect should be taken into account when synthesizing new BCD for application in PDT’ should be better demonstrated and elaborated. Furthermore, the described BCD compounds are claimed to be promising photosensitizers for PDT, however the best T1 state quantum yield is 0.21, while looking for an ‘ideal photosensitizer’ requires at least 0.5 or higher (see e.g. Metal-Based Drugs, 2008, 276109, 10.1155/2008/276109 and the references within), so this should be discussed.

In the section ‘Materials and methods’ there should be more details about the BCD compounds used in this study. For example, if they were synthesized and characterized previously, at least relevant references should be provided where these details can be found. What concentrations of four BCD in 1-propanol were used?

Line 36 – missing word in: ‘increasing of the ?? of CD’

Line 44 – correct to: radical species

Line 88 – correct to: 1-propanol

Comments on the Quality of English Language

English mostly fine, only minor editing and check for typos.

Author Response

Answer to Reviewer 1

Dear Reviewer,

We thank you for your careful and competent analysis of our paper and useful comments.

In accordance with your comments we have to answer:

Comment 1:

While this paper fits well into the Quantum Technologies section of the journal, it should show greater importance for the technology it relates to. The final claim in the abstract and the conclusions, that the described ‘effect should be taken into account when synthesizing new BCD for application in PDT’ should be better demonstrated and elaborated.

Answer:

We have tried to make our conclusion more valid by including the following statement:

“One can expect that BCD with the central heterocycle possessing major efficiency of the electron tunneling will possess higher jT. This aspect is very promising for the synthesis of new effective PS based on cyanine dyes with two interacting chromophores.”

More detailed analysis of this aspect is in the text of discussion.

Comment 2:

The described BCD compounds are claimed to be promising photosensitizers for PDT, however the best T1 state quantum yield is 0.21, while looking for an ‘ideal photosensitizer’ requires at least 0.5 or higher.

Answer:

Generally, just compounds with the triplet state quantum yield > 50% are considered as promising for PDT (See ref. [32]). However, it is valid for PS molecules with rigid structures, such as porphyrin like compounds. The values for these compounds demonstrate low dependence on the molecule environment, while for CD jT increases dramatically at the dye binding with biological structures.

We have already shown that the BCD1 demonstrates photocytotoxicity toward cancer cells higher than Photofrin, the compound, applied nowadays in clinics as PS in the PDT cancer treatment (See ref. [13]), independently that BCD1 jT is much lower than that of Photofrin. It was shown that its penetration into the cell interior is much faster than that of Photofrin ([13]). Thus, we can consider BCD as a promising PS for PDT.

We expect that BCD jT increases at binding with cell structures. This could explain its photocytotoxicity higher than that of Photofrin. This supposition is in accordance with the fact that the quantum yield of the BCD1 fluorescence increases more than 20 times (from 0.02 to 0.44) (See ref. [31]) when the dye is bound with DNA. Moreover, one can expect that BCD with the central heterocycle possessing major efficiency of the electron tunneling will possess higher jT. This aspect is very promising from the point of view of the synthesis of new effective PS, but it needs more detailed studies.

This discussion we have included in the manuscript text.

Comment 3:

In the section ‘Materials and methods’ there should be more details about the BCD compounds used in this study. For example, if they were synthesized and characterized previously, at least relevant references should be provided where these details can be found.

Answer:

We have included the respective information to the manuscript.

Comment 4:

Line 36 – missing word in: ‘increasing of the ?? of CD’

Answer:

The correction was made.

Comment 5:

Line 44 – correct to: radical species

Answer:

The correction was made.

Comment 6:

Line 88 – correct to: 1-propanol

Answer:

The correction was made.

Reviewer 2 Report

Comments and Suggestions for Authors

Borissevitch and co-worker present in their submission to Technologies “Characterization of triplet state of cyanine dyes with two chromophores. Effect of molecule structure”. The major issue is that the referee cannot follow the definition of the angles between the two chromophores, which is one of the bases for the whole article. All issues mentioned below must be carefully addressed during revision. The revised manuscript must be checked again before eventually publication can proceed.

“The solutions of four bichromophoric cyanine dyes (BCD, Figure 1)” Since BCD are known compounds, the corresponding references must be cited here (e. g. but not limited to: F. Zimmer, Journal fur praktische Chemie, 1979, vol. 321; Borisevich et al., Doklady Physical Chemistry, 1977, vol. 232-234).

Figure 1: The Lewis drawings are poorly done and must be improved. sp2 hybridized carbon atoms must be drawn as such, not as sp1 ones. The C=C double bonds must be clearly indicated as either cis (or more likely trans).

Figure 1 and Table 1: where do the angles come from? For BCD2 one would expect also an 180° angle, whereas for BCD3 one would expect an angle much bigger than 90°.

Figure 2: The data from Figure 2a (e. g. 4.5 mJ and delta A = 0.033) cannot be found in Figure 2b.

Legend of Figure 2: The wavelength must be indicated for 2a.

“and the electron tunneling through the central heterocycle, which binds the chromophores,” should be “and the electron tunneling through the central heterocycle, which connects the chromophores,”

References:

The journal names must be written consistently (either abbreviated or not).

Ref. 18: the journal name is missing, the editor information (“Edited by J.R. Winkler, California Institute 400 of Technology, Pasadena, CA, accepted by Editorial Board Member H.B. Gray”) must be deleted.

Comments on the Quality of English Language

The English must be improved. Some (but not all!) problematic sentences are mentioned in the previous section.

Author Response

Answer to Reviewer 2

Dear Reviewer,

We thank you for your careful and profound analysis of our paper and important comments.

In accordance with your comments we have to answer:

Comment 1:

The major issue is that the referee cannot follow the definition of the angles between the two chromophores, which is one of the bases for the whole article. All issues mentioned below must be carefully addressed during revision. Answer: Our results demonstrate that there is no correlation between the BCD triplet state characteristics and the angle between chromophores. Therefore we cannot consider angle as the bases of the whole article. However, we have included in Discussion the information about the relationship between the BCD spectroscopic chraracteristics and the angle between chromophores (See equation (11)). Comment 2:“The solutions of four bichromophoric cyanine dyes (BCD, Figure 1)” Since BCD are known compounds, the corresponding references must be cited here (e. g. but not limited to: F. Zimmer, Journal fur praktische Chemie, 1979, vol. 321; Borisevich et al., Doklady Physical Chemistry, 1977, vol. 232-234). Answer:We have included various basic references related with bichromophoric cyanine dyes to the list of References. Comment 3:

Figure 1: The Lewis drawings are poorly done and must be improved. sp2 hybridized carbon atoms must be drawn as such, not as sp1 ones. The C=C double bonds must be clearly indicated as either cis (or more likely trans).

Figure 1 and Table 1: where do the angles come from? For BCD2 one would expect also an 180° angle, whereas for BCD3 one would expect an angle much bigger than 90°.

 Answer:The dye structures shown in Figure 1 have just an illustrative function and cannot be used for determination of the angle between chromophores. This angle was determined from the spectroscopic data in accordance with the equation (11). However, in accordance with your recommedations we have changed these structures for those of better quality. Comment 4:

Figure 2: The data from Figure 2a (e. g. 4.5 mJ and delta A = 0.033) cannot be found in Figure 2b.

Legend of Figure 2: The wavelength must be indicated for 2a. Answer:The Figure 2 was corrected in accordance with your recommendations. Comment 5:

“and the electron tunneling through the central heterocycle, which binds the chromophores,” should be “and the electron tunneling through the central heterocycle, which connects the chromophores,”

Answer:The correction was made. Comment 6:The journal names must be written consistently (either abbreviated or not). Answer:We have formated the references in accordance with the journal rules. Comment 7:

Ref. 18: the journal name is missing, the editor information (“Edited by J.R. Winkler, California Institute 400 of Technology, Pasadena, CA, accepted by Editorial Board Member H.B. Gray”) must be deleted.

Answer:Ref. 18 is now Ref. 23. It was corrected. The spell-check was made and the text was corrected by a specialist in the area, native English.

Reviewer 3 Report

Comments and Suggestions for Authors

This report relates to photophysical properties of a series of cyanine dyes with a view toward their possible role in photodynamic therapy. In addition to these properties, a successful agent will need to [1] preferentially localize in sensitive sites in neoplastic cells, [2] have an appropriate absorbance spectrum so as to use wavelengths of light to which tissues are relatively transparent, and [3] perhaps initiate other in vivo responses known to be associated with successful PDT efficacy, e.g., vascular shutdown and initiation of an immune response. Absorbance spectra shown in Ref. 8 indicate some of the agents described have no absorbance at wavelengths < 600 nm and therefore would be of limited use in photodynamic treatment. 

While effective conversion of oxygen into reactive species is important, other factors are important regarding PDT efficacy in protocols intended for clinical applications.  The principles set forth in this report may have general applicability. This version is listed as ‘v2' and a ‘v1' version is included but the difference between these versions is not obvious and it is not clear whether this report had been reviewed previously.

How it is concluded (Abstract) that these are ‘promising photosensitizers for photodynamic therapy’ is unclear since no absorbance spectra are shown, there is no evidence for selective accumulation in neoplastic loci and no evidence of accumulation at intracellular loci where irradiation will result in initiation of death pathways.  In order to promote the concept that the increasing quantum yields of T1 might be important, is there any evidence that this fills an unmet need? Is there evidence that yields of reactive oxygen species in the context of PDT are inadequate? If the authors are going to claim that this work somehow bears on the efficacy of photodynamic therapy, this is not established. It might be better to omit PDT from the discussion if no evidence can be cited. There are only minor errors in the text, e.g., the incorrect spelling of propanol on line 88

Comments on the Quality of English Language

Adequate

Author Response

Answer to Reviewer 3

Dear Reviewer,

We thank you for your careful and competent analysis of our paper and important comments, useful for discussion.

In accordance with your comments we have to answer:

Comments:

This report relates to photophysical properties of a series of cyanine dyes with a view toward their possible role in photodynamic therapy. In addition to these properties, a successful agent will need to [1] preferentially localize in sensitive sites in neoplastic cells, [2] have an appropriate absorbance spectrum so as to use wavelengths of light to which tissues are relatively transparent, and [3] perhaps initiate other in vivo responses known to be associated with successful PDT efficacy, e.g., vascular shutdown and initiation of an immune response. Absorbance spectra shown in Ref. 8 indicate some of the agents described have no absorbance at wavelengths < 600 nm and therefore would be of limited use in photodynamic treatment. 

 While effective conversion of oxygen into reactive species is important, other factors are important regarding PDT efficacy in protocols intended for clinical applications.  The principles set forth in this report may have general applicability. This version is listed as ‘v2' and a ‘v1' version is included but the difference between these versions is not obvious and it is not clear whether this report had been reviewed previously.

 How it is concluded (Abstract) that these are ‘promising photosensitizers for photodynamic therapy’ is unclear since no absorbance spectra are shown, there is no evidence for selective accumulation in neoplastic loci and no evidence of accumulation at intracellular loci where irradiation will result in initiation of death pathways.  In order to promote the concept that the increasing quantum yields of T1 might be important, is there any evidence that this fills an unmet need? Is there evidence that yields of reactive oxygen species in the context of PDT are inadequate? If the authors are going to claim that this work somehow bears on the efficacy of photodynamic therapy, this is not established. It might be better to omit PDT from the discussion if no evidence can be cited. There are only minor errors in the text, e.g., the incorrect spelling of propanol on line 88

Answer:

The purpose of this study was an attempt to establish relationship between the structure of cyanine dye molecules with two interacting chromophores and the characteristics of their first triplet state T1, namely, its quantum yield (jT) and energy (ET).

We mention PDT only as one of their possible and important applications. However, we can already state that these dyes have properties that make them promising for PDT.  1.     All the studied dyes possess intensive peaks of optical absorption spectra in the range  > 600 nm (see Table 2 and reference [10] Fig.5(a)) 2.     BCD possess high affinity with biological structures, DNA, in particular (See ref. [31])

 3.     We have already shown that the BCD1 demonstrates photocytotoxicity toward cancer cells higher than Photofrin, compound, applied nowadays in clinics as the PS in the PDT cancer treatment (See ref. [13]), independently that BCD1 jT is much lower than that of Photofrin.  4.     It was shown that its penetration into the cell interior is much faster than that of Photofrin [13].

 5.     We have analyzed the BCD localization in the cell structure, demonstrating that mitocondrias are the principal cites (See ref. [13])  Moreover, we have already established the mechanism of the BCD phototoxicity. Now these compounds are being testing on experimental animals, demonstrating promising results. Thus, we have reason to call these compounds promising for PDT Ð¢hese studies are carried out by a large team of researchers and for the moment we have no right to present their results in this article, especially since they are not related to the topic of this article.  We did not understand your statement: “The principles set forth in this report may have general applicability. This version is listed as ‘v2' and a ‘v1' version is included but the difference between these versions is not obvious and it is not clear whether this report had been reviewed previously”. What versions V2 and V1 are you mentioning?  There are only minor errors in the text, e.g., the incorrect spelling of propanol on line 88  The correction was made.

Round 2

Reviewer 1 Report

Comments and Suggestions for Authors

The authors have answered the questions raised, and revised the manuscript accordingly.

Comments on the Quality of English Language

The newly added text contains minor grammatical and spelling errors.

Author Response

Answer to Reviewer 1

Dear Reviewer,

We thank you for your positive еvaluation of our paper.

In accordance with your comment, we have to answer:

Comment:

The newly added text contains minor grammatical and spelling errors.

Answer:

We have tried to correct all the erros in the text.

Reviewer 2 Report

Comments and Suggestions for Authors

Borissevitch and co-worker have improved their submission to Technologies “Characterization of triplet state of cyanine dyes with two chromophores. Effect of molecule structure”. Unfortunately, a number of old issues are still persisting despite the authors confirmations that they addressed them. Since the authors were not able to do a proper revision, the next revision must be checked again before eventually publication can proceed.

“in 1-proranol from Sigma – Aldrich Company” should be “in 1-propanol from Sigma – Aldrich Company”

Figure 1: The Lewis drawings, though improved, are still unacceptably poor. The authors must use a decent software for chemical drawing (there is even freeware like AC/Labs ChemSketch) and redraw all structures instead of copying the poor structures from reference 10 without properly acknowledging the copyright.

Table 1: As written in Table 2 of ref. 10, the alpha angles are not exactly 90, 120 ,150 or 180 (why should they?), therefore the proper experimental angles should be mentioned.

Figure 2: The data from Figure 2a (e. g. for 0.5, 0.7 and 3.2 mJ STILL cannot be found in Figure 2b.

Despite the authors statement “We have formated the references in accordance with the journal rules.”, the references are still NOT consistently formatted (see e. g. “Dyes and Pigments” and “Chem Sci.”).

Author Response

Answer to Reviewer 2

Dear Reviewer,

We thank you for your scrupulous and competent analysis of our paper.

In accordance with your comments, we have to answer:

Comment 1:

“in 1-proranol from Sigma – Aldrich Company” should be “in 1-propanol from Sigma – Aldrich Company”

Answer:

The correction was made.

Comment 2:

Figure 1: The Lewis drawings, though improved, are still unacceptably poor. The authors must use a decent software for chemical drawing (there is even freeware like AC/Labs ChemSketch) and redraw all structures instead of copying the poor structures from reference 10 without properly acknowledging the copyright.

Answer:

The BCD structures were not copied from the reference 10 but prepared using the official program ChemWindows widely applied especially for this purpose. Anyway, in accordance with your recommendation we have prepared a new figure 1 using Marvin Sketch 21.17 program and copied with the help of ChemDraw 17.0 program. We hope this form of presentation of the BCD structures be more acceptable.

Comment 3:

Table 1: As written in Table 2 of ref. 10, the alpha angles are not exactly 90, 120 ,150 or 180 (why should they?), therefore the proper experimental angles should be mentioned.

Answer:

In accordance with your recommendation, we have included into Table 2 the angles between chromophores, calculated by the equation (11) using the spectroscopic data from the table. We must clarify that the angles between chromophores vary slightly in different solvents. The angles given in Ref. 10 were determined in ethanol. In the present work, 1-propanol was used as a solvent. Therefore, the obtained values of the angles differ slightly from those given in Ref. 10.

Comment 4:

Figure 2: The data from Figure 2a (e. g. for 0.5, 0.7 and 3.2 mJ STILL cannot be found in Figure 2b.

Answer:

Figure 2a shows the kinetic curves for TPPS4. All the data you listed are presented in Figure 2b:E = 0.5 mJ  DeltaA = 0.01; E = 0.7 mJ DeltaA = 0.017; E = 3.2 mJ DeltaA = 0.047.Тo eliminate further confusion, we have modified Figure 2a and included all the points from the Figure 2b related to TTPS4 to Figure 2a.

Comment 5:

Despite the authors statement “We have formated the references in accordance with the journal rules.”, the references are still NOT consistently formatted (see e. g. “Dyes and Pigments” and “Chem Sci.”).

Answer:

The corrections were made.

Round 3

Reviewer 2 Report

Comments and Suggestions for Authors

Borissevitch and co-worker have further improved their submission to Technologies “Characterization of triplet state of cyanine dyes with two chromophores. Effect of molecule structure”. One detail needs to be fixed and then publication can proceed.

Figure 1: The Lewis drawings are now finally drawn with a decent software. The authors must still further improve the drawing and show the sp2 hybridized carbon atoms as such (with angles between the substituents of approximately 120° each).

Comments on the Quality of English Language

OK

Author Response

Answer to Reviewer 2

Dear Reviewer,

We thank you for your work in improving the accuracy and quality of our article.

In accordance with your comment, we have to answer:

Comment:

The Lewis drawings are now finally drawn with a decent software. The authors must still further improve the drawing and show the sp2 hybridized carbon atoms as such (with angles between the substituents of approximately 120° each).

Answer:

We tried to correct the structures of the dyes in accordance with your recommendations using the program ChemSketch Freeware. We hope that this form of presentation is in accordance with the requirements.